# Trends of Total Knee Arthroplasty According to Age Structural Changes in Korea from 2011 to 2018

**DOI:** 10.3390/ijerph182413397

**Published:** 2021-12-20

**Authors:** Yong-Beom Kim, Hyung-Suk Choi, Eun Myeong Kang, Suyeon Park, Gi-Won Seo, Dong-Il Chun, Tae-Hong Min

**Affiliations:** 1Department of Orthopedic Surgery, Soon Chun Hyang University Seoul Hospital, Seoul 04401, Korea; schkyb@schmc.ac.kr (Y.-B.K.); knee@schmc.ac.kr (H.-S.C.); 129741@schmc.ac.kr (E.M.K.); 102980@schmc.ac.kr (G.-W.S.); orthochun@gmail.com (D.-I.C.); 2Department of Biostatistics, Soonchunhyang University Seoul Hospital, Seoul 04401, Korea; suyeon1002@schmc.ac.kr; 3Department of Applied Statsitics, Chung-Ang University, Seoul 06911, Korea

**Keywords:** Total Knee Arthroplasty (TKA), Health Insurance Review and Assessment Service (HIRA), National Health Insurance Service (NHIS), Korean Statistical Information Service (KOSIS), trend, age

## Abstract

Total Knee Arthroplasty (TKA) is one of the most commonly performed surgeries worldwide since it can improve pain, quality of life, and functional outcome. Due to the expansion of hospitals specialized in joint surgery, the topography of TKA implementation in Korea is changing. This study analyzed longitudinal trends of TKA based on changes in age distribution, sex, hospital, and region based on the Health Insurance Review and Assessment Service (HIRA) of Korea database. Data were collected from the National Health Insurance Service (NHIS), the Korean Statistical Information Service (KOSIS), and the Health Insurance Review and Assessment Service (HIRA) in Korea for the period 2011–2018. Results show the total number of surgeries increased and the number of patients by age decreased in those under the age of 70, while the number of patients over 70 years of age increased. A remarkable increase in women was found, and there was no significant difference between regions. TKA is spreading in a more universal and easily accessible form in Korea and has increased more in other relatively small medical institutions compared to tertiary referral medical centers. Due to the increase of orthopedics’ specialized hospitals and clinics, TKA is becoming more prominent in those hospitals.

## 1. Introduction

The prevalence of knee osteoarthritis has been increasing in recent years [1,2,3]. Among the treatment of end stage osteoarthritis, Total Knee Arthroplasty (TKA) is one of the most effective treatments and it is one of the most commonly performed surgeries worldwide since it can improve pain, quality of life, and functional outcome [4,5].

As population aging is progressing not only in the West, but also in East Asia, total volume of primary knee arthroplasty has increased steadily over the past few decades [5,6,7]. Kurtz et al. predicted that the annual number of TKA implementation in the U.S. will increase from 1.37 million in 2020 to 3.48 million in 2030 [8]. The trends of TKA use are correlated with prevalence of knee osteoarthritis, and is mainly influenced by changes in demographic factors such as age, gender, life expectancy [1,2,5,9]. Since predicted volume of overall medical cost for TKA surgery and its management will increase over the next several decades, an accurate analysis of TKA epidemiology is required for better public health care policy [10].

Previous studies regarding epidemiology of TKA in Western countries presented trends of increasing use in elderly patients, growing proportion of younger population and female dominance [1,5,7,11,12,13,14,15]. In Eastern Asian epidemiologic studies, similar trends were observed with growth of use in the aging population and a marked greater use in female patients [16,17,18]. However, previous studies in Eastern Asian population have simply measured the rate of TKA implementation according to single demographic factors in a cross-sectional manner without taking into account the trend of change in the population by age [18,19]. Since Korea is now facing the issue of an aging society, changes in age structural change are expected [20]. There is a need for research on changes in the TKA implementation rate according to changes in the population structure [9].

Moreover, there is a growing discrepancy in demographic structure in the Korean population, with about half of the total population living in the metropolitan areas, including Seoul and Gyeonggi-do, while populations in other rural regions are shrinking [20,21,22]. In addition, due to the expansion of clinics specialized in orthopaedic joint surgery, the topography of TKA implementation in Korea is changing. Accordingly, it is necessary to compare trends on TKA implementation rates by these changing social demographic factors.

Most previous epidemiologic studies have been performed based on a single center or multi-center. Few reports have analyzed TKA trends in a large claim database [18]. To evaluate representative trends, a nationwide population should be analyzed [22].

Therefore, this study aims to analyze longitudinal trends of TKA based on changes in age distribution, sex, hospital, and region based on Health Insurance Review and Assessment Service (HIRA) of Korea database.

## 2. Materials and Methods

### 2.1. Data Collection

Data were collected from the National Health Insurance Service (NHIS, Wonju-si, Korea), the Korean Statistical Information Service (KOSIS, Sitemap, Seoul, Korea), and the Health Insurance Review and Assessment Service (HIRA, Wonju-si, Korea) in Korea for the period 2011–2018. In particular, HIRA supplied data after de-identification, which encompassed age, sex, diagnosis, hospital visit dates, medication prescribed in both inpatient and outpatient visits, hospitalization, medical interventions, and visits to the emergency department. For the purpose of research sample identification, a number of codes (TKA, N2072, and N2077) were employed for Total Knee Arthroplasty. Meanwhile, data obtained from KOSIS included information on age, sex, hospital, and region.

### 2.2. Statistical Analysis

An annual estimation of cumulative patients was conducted. All results were approximated based on the formula number of patients (for the year) × 100,000/total population (for the year). As a primary outcome, this study aimed to establish the prevalence trend per year and whether the factors of age, sex, hospital, and region were associated with distinct patterns. To that end, patients were separated into four age groups: younger than 60, 60s, 70s, and older than 80. Regarding hospitals, they were distinguished into tertiary referral center, general hospital, orthopaedic surgery clinic. Tertiary referral center is defined as university hospital with an approved residency program. General hospital is a hospital with medical departments but no residency program. Orthopaedic surgery clinics are institutions specialized in orthopaedic surgery but without other medical departments. Regarding areas, three groups were differentiated: Seoul and Gyeonggi, and other regions.

Data were processed via Poisson regression analysis alongside two models. Model 1 involved calculation of the effect size of the general time trend, while model 2 involved investigation of potential inter-group discrepancies in time trend estimates based on integration of the interaction effect to model 1. Expression of results took the form of incidence rate ratio (IRR) and 95% confidence interval (CI), with variables of age, sex, hospital, and region being used to characterize patient rate per 100,000 population. Furthermore, Spearman correlation analysis was performed to quantify and analyze how the patient rate per 100,000 population was correlated with the proportion of the population. A two-sided test was carried out, with statistical significance considered when *p*-value was less than 0.05. R software Version 3.6.3 (R Foundation, Vienna, Austria) was employed for the entirety of data analysis.

## 3. Results

Over the 2011–2018 research period, the cumulative patient number increased from 44,361 to 64,456 and the patient rate per 100,000 people increased from 44.3 to 62.8 (time trend = 1.05; 95% CI: 1.04–1.06) irrespective of the population (Figure 1). An explanation for these observations is provided in Table 1 and Table 2. Model 1 showed an increasing trend in the positive direction except for age adjustment.

The Age × Year of model 2 revealed statistically significant pattern differences between age groups. However, the general trend in model 1 did not show statistically significant difference (time trend = 0.99; 95% CI: 0.97–1.00). In model 2, with the group under 60 years as a reference group, the number of patients per 100,000 people decreased from 4.8 to 3.6. In the 60s group, the number of patients per 100,000 people declined more from 216.8 to 195.5. The difference in slope was statistically significant (time trend difference = 0.97; 95% CI: 0.95–0.99). The number of patients per 100,000 people rose slightly from 118.9 to 208.2 in the 80s age group and older (time trend difference = 1.04; 95% CI: 1.02–1.06), with the highest increase in the 70s age group (from 376.6 to 476.5, time trend difference = 1.04; 95% CI: 1.02–1.06).

Medical services were found mainly in Seoul, which was reflected in the fact that the number of patients rose quickly in Seoul. In terms of area, results were statistically significant in model 1 (time trend = 1.05; 95% CI: 1.05–1.06). Model 2 yielded identical outcomes as model 1, showing a rise from 130.1 to 175.1 in Seoul, from 56.4 to 84.1 in Gyeonggi, and from 87.1 to 127.7 in others. However, there were no statistically significant differences in time trend between others and Seoul (time trend difference = 1.00; 95% CI: 0.99–1.02).

In model 2, gender trends tended to increase significantly in women over men (time trend difference = 1.09; 95% CI: 1.09–1.10) with the number of patients per 100,000 people ranging from 77.7 to 104.9 for women and from 10.9 to 20.6 for men.

The hospital x year of model 2 findings also revealed an increase in the percentage of patients per 100,000 people for all types of medical centres considered (from 5.4 to 7.3 for tertiary referral center, from 10.7 to 18 for general hospitals, and from 28.2 to 37.5 for others). Comparison between the time trend and tertiary referral center showed a significant difference in general hospital (time trend difference = 1.04; 95% CI: 1.02–1.06). The clinic group also presented a statistically significance (time trend difference = 1.05; 95% CI: 1.04–1.05).

The association between the proportion of patients per 100,000 people and the proportion of population by year for every age group is illustrated in Figure 2.

The purpose of this comparison was to establish how the patient ratio was correlated with the population ratio rise and fall for each age group and year. Apart from the 70s age group and the over 80 age group, there was a reduction in the patient rate per 100,000 people for every age group. Furthermore, for the age group of over 60 year olds, there was an increase in population proportion according to age group and year, whereas for the age group of under 60 years old, this proportion declined. This observation can be attributed to aging. Moreover, regarding the association between patient rate and population proportion by year, the 60s age group showed a negative association (r = −0.519. *p* = 0.187), while other age groups showed positive associations (r = 0.884, *p* = 0.004 for under 60s; r = 0.827, *p* = 0.011 for 70–79; r = 0.979, *p* < 0.001 for over 80 s).

## 4. Discussion

Various studies have reported epidemiology and projection of TKA in western countries [6,7,23]. However, few authors have investigated trends of TKA in the Asian population [11,16,24]. Previous epidemiologic studies have mainly investigated demographic factors such as duration of disease, age, body mass index, level of education, and disease grade [3,18,24,25].

South Korea is now facing the issue of an extreme aging society among Organization for Economic Cooperation and Development (OECD) countries. It is expected to show structural changes in age distribution [20,26]. This is the first study to analyze nationwide epidemiology of TKA in Korea reflecting age structural changes.

In Korea, there is a difference between urban and rural areas in terms of access to medical care [20]. In addition, TKA is widely implemented in hospitals of various levels due to the spread of the medical insurance system. However, reports investigating the discrepancy between regions or hospital levels have not been reported. To the best of our knowledge, this is the first study to report regional and hospital level variances in TKA epidemiology. Through this analysis, more efficient allocation of medical resources can be achieved.

First, from 2011 to 2018, the number of TKA implementations per 100,000 people increased, indicating that the actual total volume of TKA increased. However, our findings showed that the number of patients undergoing TKA decreased in patients under the age of 60 as a general trend. This differed from previous studies reporting an increased proportion of younger patients [16,27]. Since this has a positive relation, it can be inferred that the number of TKAs has decreased due to the aging of the demographic structure as the number of people under the age of 60 decreases.

Interestingly, for the age group between 60 and 69 years, the proportion of the population distribution increased over time, but the number of patients undergoing TKA tended to decrease. The reason for this negative relation seems to be the expansion of joint preserving surgeries such as high tibial osteotomy and unicompartment knee arthroplasty [28,29,30,31]. In addition, the expansion of conservative treatments such as stem cell therapy, injection therapy, and drug therapy might have led to the higher TKA in this age group than before [32,33,34,35]. Furthermore, considering that the lifespan of a conventional artificial joint replacement implant is about 15–20 years and that the average lifespan of Koreans has recently exceeded 80 years, those in their 60s might prefer joint preserving surgery to artificial joints [8,15,36].

The number of TKA is also increasing in the patient group with age over 70 years old. At the same time, the proportion of population distribution is also increasing. It can be inferred that this is also due to the aging society.

In terms of sex discrepancy, similar to previous studies, our study showed consistent female dominance [18,19]. Both men and women showed increased rates of TKA. However, higher rates of increase were observed in women. This is because osteoarthritis of the knee joint is more prevalent in women.

In terms of regional differences, there was an increase in the total number of TKA in all regions. Among them, the Gyeonggi area showed a relatively lower increase in the total number of TKA compared to Seoul. However, the increase was statistically significant. Although the area of others showed the greatest increase in the number of TKA, the increase was statistically insignificant. From these results, it was found that there was no significant regional differences in the implementation of TKA in Korea, indicating that there was no difference in regional accessibility to TKA surgery in Korea. This result contradicted a previous study by Jain et al. [11], the TKA implementation rate in the United States has regional variance according to time period. The implementation rate of TKA surgery tended to increase between 1990–2000 in urban area while rural area tended to decrease.

Lastly, the number of TKA trials according to time between grades in hospitals changed significantly. This suggests that TKA is spreading in a more universal and easily accessible form in Korea than before in that it has increased more in other relatively small medical institutions than in tertiary referral medical centers. In addition, due to the increase of orthopaedics specialized surgery clinics, TKA is receiving more prominent attention in those hospitals.

However, our research also has limitations. First, there was no comparison of body mass index, socioeconomic status, activity level, or lifestyle. However, the purpose of this study was to perform a macroscopic analysis of the longitudinal trend of TKA across the country. Thus, many individual patient factors had to be excluded.

Second, a detailed analysis of the region was not conducted. Due to the nature of the Korean social climate, surgery is often performed in the capital city, not in the patient’s residential area. The data of these patients were not properly reflected. In addition, the implementation of TKA in regional metropolitan areas, such as Busan and Gyeongnam, was not properly reflected. Therefore, more specified regional analysis is required in future studies.

Thirdly, in Korea, the Korean institute for healthcare accreditation (KOIHA) regularly conducts medical institution certification evaluations to determine hospital level. Grades of some medical institutions had some modification due to this certification evaluation from 2011 to 2018, which might have acted as a bias for this study.

Lastly, the comparison of this paper was made based on the number of patients. However, patients with TKA might have both knees affected, which is different from the actual number of cases. Further study on this is necessary.

Despite these limitations, this study is still meaningful in that it analyzes national longitudinal trends for the last 10 years. In addition, based on the results of this study, we can predict the socioeconomic burden due to TKA in future. According to a previous study on nationwide German healthcare by Klug et al. [37], it is predicted that the socioeconomic burden caused by TKA will increase by 43% over the next 30 years. As the burden of artificial joint surgery is increasing worldwide, this study will be helpful in applying age-specific and selective welfare strategies while establishing a nationwide healthcare system.

## 5. Conclusions

This study analyzed the trend of Total Knee Arthroplasty in Korea from 2011 to 2018. Yearly, the total number of surgeries increased, the number of patients by age decreased in those under the age of 70, while the number of patients over 70 years of age increased. Particularly in the 60s age group, the number of patients showed a characteristic tendency to decrease compared to the increasing population ratio. A remarkable increase was evident in women. Increases were seen across all hospital levels; however, compared to tertiary referral centers, it increased further at the clinic level. By region, it was most frequently implemented in Seoul. However, there was no significant difference between regions.

## Figures and Tables

**Figure 1 ijerph-18-13397-f001:**
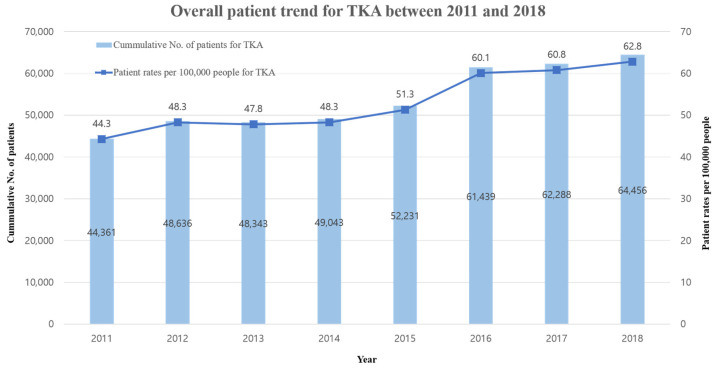
Overall patient trend for TKA between 2011 and 2018.

**Figure 2 ijerph-18-13397-f002:**
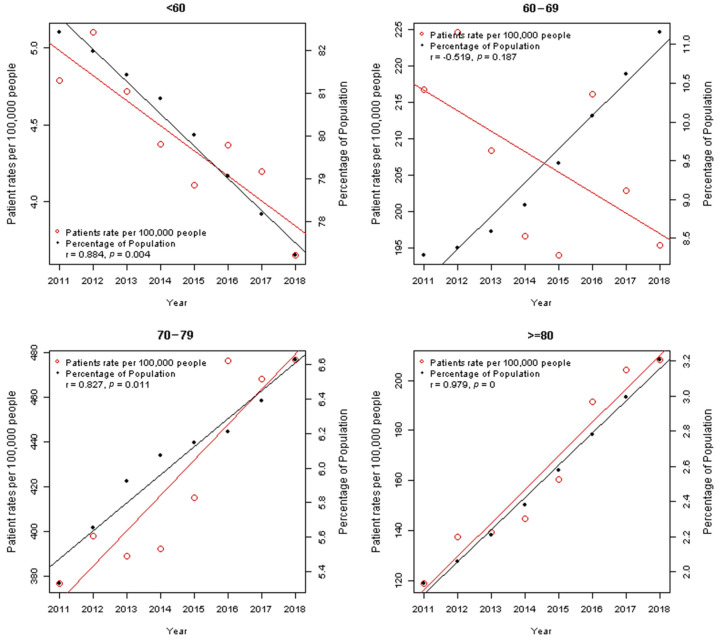
The association between the proportion of patients per 100,000 people and the proportion of population by year for every age group.

**Table 1 ijerph-18-13397-t001:** Model 1—Poisson regression analysis according to calculation of the effect size of the general time trend.

Year ^1^									Time Trend		
TKA—Model 1	2011	2012	2013	2014	2015	2016	2017	2018	IRR ^2^	95%CI	
										Lower	Upper
All	44.3	48.3	47.8	48.3	51.3	60.1	60.8	62.8	1.05	1.04	1.06
Year with Age Adjustment									0.99	0.97	1.00
Year with Sex Adjustment									1.05	1.04	1.06
Year with Hospital Adjustment									1.05	1.04	1.06
Year with Area Adjustment									1.05	1.05	1.06

^1^ unit: patient rate per 100,000 people. ^2^ IRR: incidence rate ratio.

**Table 2 ijerph-18-13397-t002:** Model 2—Poisson regression analysis according to investigation of potential inter-group discrepancies in time trend estimates.

Year ^1^										Time Trend		
TKA—Model 2		2011	2012	2013	2014	2015	2016	2017	2018	IRR ^2^	95%CI	
											Lower	Upper
Year × Age ^3^	<60	4.8	5.1	4.7	4.4	4.1	4.4	4.2	3.6	Reference		
	60–69	216.8	224.6	208.4	196.6	194.1	216.2	203	195.5	0.97	0.95	0.99
	70–79	376.6	397.9	388.6	392.3	415	476.2	468	476.5	1.04	1.02	1.06
	≥80	118.9	137.5	139.5	144.9	160.4	191.5	204.3	208.2	1.04	1.02	1.06
Year × Sex ^3^	Male	10.9	12.2	13.1	13.7	14.6	17.7	18.5	20.6	Reference		
	Female	77.7	84.5	82.5	82.9	87.9	102.4	102.9	104.9	1.09	1.09	1.10
Year × Hospital ^3^	Tertiary	5.4	5.8	5.6	6.3	6.6	8	6.6	7.3	Reference		
	General	10.7	10.9	11.9	12.1	13.6	16.3	16.7	18	1.04	1.02	1.06
	Clinic	28.2	31.7	30.4	29.9	31.1	35.8	37.6	37.5	1.05	1.04	1.05
Year × Area ^3^	Seoul	130.1	139.1	131.4	135.6	139	168.2	167.1	175.1	Reference		
	Gyeonggi	56.4	62.4	64	67.1	70.5	76.4	81.1	84.1	0.99	0.98	1.00
	Others	87.1	96	96.5	95.7	103.7	123	124.1	127.7	1.00	0.99	1.02

^1^ unit: patient rate per 100,000 people. ^2^ IRR: incidence rate ratio. ^3^ This is the result of the interaction term from Model 2.

## Data Availability

The data are distributed to registered users through the official website of HIRA Healthcare Bigdata Hub (https://opendata.hira.or.kr/home.do, accessed on 24 August 2020). After the evaluation of a research proposal by the HIRA review committee, registered users can receive special access privileges to the data.

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
