# Peer review of "Trends of Total Knee Arthroplasty According to Age Structural Changes in Korea from 2011 to 2018"

_ijerph, 2021, doi:10.3390/ijerph182413397_

Round 1

Reviewer 1 Report

With pleasure I read the manuscript entitled “Trends of Total Knee Arthroplasty According to Age Structural Changes in Korea From 2011 to 2018” by Yong-Beom Kim et al., dealing with the important argument regarding the epidemiologic dataset concerning total knee arthroplasty (TKA) performed in Korean population and based on specific factors such as age, sex, hospital and region. The authors conclude the total number of TKA increased, the number of patients under 70 decreased, while those over 70 increased. Gender showed a significant increase in women and there was no significant difference between regions.

The manuscript is written and articulated well, in quite concise and comprehensive way. The study is supported by sufficient references and results are supported by clearly exposed data. The authors stated the limitation of the study. In my opinion the manuscript is suitable for the publication in the present form, I suggest only some minor spell check. 

Author Response

  • We authors are grateful for your generous consideration on our work. And we are looking forward to our manuscript can be published in this honorable journal safely. Thank you.

Sincerely

Corresponding author: Tae-Hong Min M.D 

Reviewer 2 Report

This is an excellent example of public health manuscript reflecting an option of resources management and destination to different geographical areas. I would change the title because misleading and reductive (it looks like only age have been considered and according to age a different surgical approach is considered)

I would also add in the discussion or in the conclusion a sentence on potential applications of this health politic approach todifferent countries 

Author Response

  • We authors are really thank you for the opinion on our work. We authors have reflected your precious advices on our revision. So we put additional sentences and references on the discussion session for better understanding and future application on public health management. in case of title, we authors think that the main difference between our study and previous one is that it had investigated the demographics and trends according to Age structural change in Korea. So with all due respect but we authors are politely asking the reviewer if we can sustain this title. 

Sincerely 

Corresponding author : Tae-Hong Min M.D

Reviewer 3 Report

Minor Comments:

  • Line 71: The authors refer to six age groups but only 4 age groups are mentioned. This needs to be corrected.
  • Line 157: A claim has been made without any reference. Please add appropriate references.

Major Comments:

  • The introduction is poorly constructed and needs to be written in a much elaborate manner.
  • Table 1 needs to be presented in a better manner. 
  • Lines 186-187: Too little information to make the conclusion and incorrect reference has been used as part of the conclusion.
  • The limitations outweigh the results and conclusions. Especially with limitations mentioned in lines 198-202, the authors cannot make conclude lines 177-187. 

Author Response

Reviewer 3

We authors are really grateful for your advice on our work. We’ve revised our manuscript according to your recommendation. And we hope our work can be published on this honorable journal.

Line 71: The authors refer to six age groups but only 4 age groups are mentioned. This needs to be corrected.

  • Thank you for the comment, and we are sorry for the mistake. We revised the manuscript

Line 157: A claim has been made without any reference. Please add appropriate references.

  • Thank you for the comment, we put additional reference for our statement.

Major Comments:

The introduction is poorly constructed and needs to be written in a much elaborate manner.

  • We authors are really grateful for your advice on our manuscript especially in introduction section. We do agree previous manuscript was not that organized enough to express our message. So we put additional statement in this section for better logical flow.

Table 1 needs to be presented in a better manner.

  • Thank you for the comment, and we revised the table 1 into more elaborate manner.

Lines 186-187: Too little information to make the conclusion and incorrect reference has been used as part of the conclusion.

  • Thank you for the comment, and we are sorry for the mistake. We put proper reference for our statement

The limitations outweigh the results and conclusions. Especially with limitations mentioned in lines 198-202, the authors cannot make conclude lines 177-187.

  • Thank you for the comment, and we do agree that second limitation is major drawback of our study. So we added statement for future work to overcome this limitation.

Sincerely 

Corresponding author: Tae-Hong Min

Round 2

Reviewer 3 Report

Minor comments have been addressed by the authors. However, the major comments still remain to be largely addressed. 

  1. The authors acknowledge the introduction has not been well written and in response have only added a couple of lines to it but overall it still largely remains same as before.  
  2. Table 1 has not been modified and is still the same. In my opinion, I feel the authors have tried to cramp a lot of data in Table 1 without proper explanation of each models in the methods and/or results section.
  3. The changes made in lines 194-197 is not sufficient. Adding a statement is not sufficient enough to add weightage to a conclusion. Also, the statement added by the authors seem to contradict the conclusion. 
  4.  The authors could add some additional details on how they plan to address the limitations with future studies. 

Author Response

1. The authors acknowledge the introduction has not been well written and in response have only added a couple of lines to it but overall it still largely remains same as before.  

  • Thank you for the comment. We do agree that there is still drawbacks in our introduction, so we re-writed and re-organized the introduction section thoroughly.

2. Table 1 has not been modified and is still the same. In my opinion, I feel the authors have tried to cramp a lot of data in Table 1 without proper explanation of each models in the methods and/or results section.

  • Thank you for the comment, We do agree that there is too many information on our table, so we separated the table into each model. And revised the result section into more explainable way.
  • Model 1 is a model that considers only risk factors(age, sex, hospital and area ) as adjustment variables as fixed effects when looking at time trends. However, model2 is a model that additionally considers the interaction term (year x risk factor).

3. The changes made in lines 194-197 is not sufficient. Adding a statement is not sufficient enough to add weightage to a conclusion. Also, the statement added by the authors seem to contradict the conclusion. 

  • Thank you for the comment, we do agree that there is some leap of logic in our statement, so we deleted our conclusion

4.  The authors could add some additional details on how they plan to address the limitations with future studies. 

  • Thank you for the comment, we stated the limitations and following future work at last portion of each paragraph in the discussion session. Most of future work should be accompanied by further analysis including detailed factors of demographic data. With all due respect, we authors think there’s no additional explanation available for the limitation.
